# Exploring the Potential of *Meyerozyma guilliermondii* on Physiological Performances and Defense Response against Fusarium Crown Rot on Durum Wheat

**DOI:** 10.3390/pathogens10010052

**Published:** 2021-01-08

**Authors:** Zayneb Kthiri, Maissa Ben Jabeur, Fadia Chairi, Camilo López-Cristoffanini, Marta López-Carbonell, Maria Dolores Serret, Jose Luis Araus, Chahine Karmous, Walid Hamada

**Affiliations:** 1Laboratory of Genetics and Plant Breeding, National Institute of Agronomy of Tunis, 43, Av Charles Nicolle, Tunis 1082, Tunisia; maissa.benjabeur@hotmail.com (M.B.J.); karmouschahine@gmail.com (C.K.); w_hamada@yahoo.com (W.H.); 2Section of Plant Physiology, University of Barcelona, 08028 Barcelona, Spain; fadia.chairi@hotmail.fr (F.C.); camilo.lopez.cr.puc@gmail.com (C.L.-C.); mlopez@ub.edu (M.L.-C.); dserret@ub.edu (M.D.S.); jaraus@ub.edu (J.L.A.); 3AGROTECNIO (Center of Research in Agrotechnology), University of Lleida, 25198 Lleida, Spain

**Keywords:** *Meyerozyma guilliermondii*, *Fusarium culmorum*, durum wheat, growth promotion, physiological traits, induced defense response

## Abstract

Coating seeds with bio-control agents is a potentially effective approach to reduce the usage of pesticides and fertilizers applied and protect the natural environment. This study evaluated the effect of seed coating with *Meyerozyma guilliermondii*, strain INAT (MT731365), on seed germination, plant growth and photosynthesis, and plant resistance against *Fusarium culmorum*, in durum wheat under controlled conditions. Compared to control plants, seed coating with *M. guilliermondii* promoted the wheat growth (shoot and roots length and biomass), and photosynthesis and transpiration traits (chlorophyll, ɸPSII, rates of photosynthesis and transpiration, etc.) together with higher nitrogen balance index (NBI) and lower flavonols and anthocyanins. At 21 days post infection with Fusarium, *M. guilliermondii* was found to reduce the disease incidence and the severity, with reduction rates reaching up to 31.2% and 30.4%, respectively, as well as to alleviate the disease damaging impact on photosynthesis and plant growth. This was associated with lower ABA, flavonols and anthocyanins, compared to infected control. A pivotal function of *M. guilliermondii* as an antagonist of *F. culmorum* and a growth promoter is discussed.

## 1. Introduction

In Tunisia, durum wheat is, by extension, the main herbaceous crop, covering over 40% of the cereal-producing areas [1]. However, durum wheat yield is affected by the harmful effects of various fungal pathogens causing drastic losses, up to 26% for durum under fusarium foot and root rot disease [2]. *F. culmorum* is the most reported Fusarium crown rot (FCR) pathogen [3,4]. The fungus infects seeds since initial growth stages leading to a decrease in seed germination and emergence rate. Moreover, the infections might occur in roots, root crown and stem tissues [5]. The control of *F. culmorum* by chemical fungicides [6] showed low efficiency due in part to the development of fungicide resistance as well as harmful environmental impact. Thus, novel and integrated management approaches against *F. culmorum* have been developed and tested [7]. In fact, seed priming using biostimulants, as seed coating agents, was used to induce germination, abscisic acid to improve emergence of seedlings [8,9,10], and to induce mechanisms of plant disease resistance [11].

Upon infection and/or treatment with biostimulants, plant often activated a highly coordinated biochemical and structural mechanisms of defence, including hormones, sugars and phenolic compounds to inhibit pathogen proliferation [12]. Besides, defence mechanisms against plant disease are built to restrain fungal pathogens through the accumulation of phenolics compounds involved in cell walls lignification [13]; The use of yeasts constitutes a new promising biostimulant approach to improve disease resistance and plant growth. These endophytic yeasts exude a range of secondary metabolites (phenols, flavonoids, abscisic acid (ABA), indoleacetic acid (IAA), etc.) to help the plant to overcome stress conditions [14].

Photosynthesis generates NADPH, ATP, and carbohydrates. These resources are used for the biosynthesis of many important compounds, such as primary metabolites, antimicrobial compounds, and defence-related hormones, such as jasmonic, salicylic and abscisic acids, ethylene, and antimicrobial compounds [15]. The key steps in the biosynthesis of defence-related hormones and their precursors take place in the chloroplast. Further, chloroplasts are major generators of reactive oxygen species (ROS) and nitric oxide, and a site for calcium signalling. These signalling molecules are important to plant defence as well [15]. However, photosynthetic activity is deeply affected under biotic stressors such as *F. verticillioides* on maize [16]. In the case of Fusarium head blight, it causes a significant reduction in the net photosynthesis rate of the flag leaf of wheat plants [17].

In recent years, yeast-induced plant resistance has been considered a potential approach to control plant pathogens; this is the case, for example, of *Pichia guilliermondii* strain Z1 against citrus blue mould on citrus [17], *P. guilliermondii* against *Rhizopus nigricans* on tomato fruit [18] and *M. guilliermondii* on rice blast, cabbage black leaf spot, and tomato bacterial wilt diseases [19]. However, to date, no reports exist on the effects of these yeasts on promoting disease tolerance in durum wheat. The present investigation constitutes the first study on the effect of seed coating with *M. guilliermondii* on plant growth of durum wheat, and on promoting defence in response to *F. culmorum* infection.

## 2. Results

### 2.1. Effect of Seed Coating with M. guilliermondii and Infection with F. culmoum on Seed Germination and Seedling Growth of Wheat

Results showed that coating with *M. guilliermondii* significantly and positively affected all measured parameters as rate of germination (*p* < 0.01), biomass (*p* ≤ 0.05), and root and shoot length (*p* ≤ 0.001) (Table 1). However, the infection with *F. culmorum* significantly affected only the root and shoot length and biomass (*p* ≤ 0.001).

Under non-infected conditions, seed coating with *M. guilliermondii* increased the germination rate of seeds up to 93.3% compared to control seeds that reached only 46.7%. Moreover, it increased the shoot and root length and plant biomass (Table 1). In control plants, the infection with *F. culmorum* induced a reduction in germination rate (DR = 42.8%), shoot length (31.7%), root length (33.8%), and in plant biomass (50.5%). In plants treated with *M. guillermondii*, the infection resulted in a lesser reduction of germination rate (DR = 24.9%), shoot length (25.9%), root length (18.4%), and of plant biomass (31.4%), compared to infected control (i.e., untreated) plants.

### 2.2. Effect of Seed Coating with M. guilliermondii on the Incidence and Severity of Fusarium Crown Rot

A significant increase over the time was showed for both the disease incidence and severity (Figure 1); basing on the symptoms caused by the endophytic fungus *F. culmorum* on wheat plants at 14, 21 and 28 das. At 14 das, we observed a slight significant difference in the disease incidence and severity (*p* < 0.05) between *M. guilliermondii*- treated plants and control plants. At 21das, the effect of *M. guilliermondii* in controlling the Fusarium crown rot development was more remarkable compared to the control plants (*p* < 0.01). The impact of the seed coating with *M. guilliermondii* was further confirmed at 28 das, depicted by significant lower incidence and severity on the crowns, compared to control (*p* < 0.01), with reduction rates reaching up to 31.18% and 30.43%, respectively (Figure 1).

### 2.3. Effect of Seed Coating with M. guilliermondii and Infection with F. culmorum on the Change in Leaf Pigments

The two-way ANOVA showed that contents of chlorophyll, flavonoids and anthocyanins and NBI of leaves were significantly affected by the coating treatment, the infection, and their interaction (Table 2). The one-way ANOVA showed that the treatments significantly affected chlorophyll, flavonols, anthocyanins, and NBI at almost all the time points (Figure 2).

In non-infected conditions, the coating of seeds resulted in higher chlorophyll content and NBI, compared to control, which was more remarkable at 28 das (Figure 2). Additionally, the coating of seeds resulted in a lower content of flavonols and anthocyanins, compared to control. Under infected conditions, control plants witnessed a continuous decrease in chlorophyll and flavonols, associated with a continuous increase of anthocyanins, and a low NBI, compared to non-infected control. Whereas, in the plants derived from infected seeds treated with *M. guilliermondii*, a continuous increase and a higher level of the chlorophyll content, compared to the infected control, was observed, remarkably at 28 das (IR = 20.9%). In addition, compared to the infected control, the seed coating treatment resulted in lower contents of flavonols (DR = 8.4%) and anthocyanins (DR = 39.6%) and a higher NBI (IR = 27.67%) in leaves.

### 2.4. Effect of Seed Coating with M. guilliermondii and Infection with F. culmorum on the Leaf Gas Exchange, Photosynthesis, Stomatal Conductance (Sc), and ABA Content

At 28 das, the results showed that both the coating and infection significantly affected the photosynthesis rate, the stomatal conductance, the transpiration rate, the electron transport rate, and the non-photochemical quenching. However, only the infection significantly affected the intercellular to atmospheric CO_2_ concentration ratio (Ci/Ca) and the quantum efficiency of Photosystem II. Moreover, the ABA content was only significantly affected by the coating treatment (Table 3). In addition, only the Ci/Ca ratio was significantly affected by the interaction coating x infection (C × I).

Under non-infected conditions, the seed coating by *M. guilliermondii* enhanced the photosynthesis rate, the stomatal conductance, the internal CO_2_ concentration, the electron transport rate, the transpiration rate, and decreased the non-photochemical quenching and the ABA content, with no significant impact on the quantum efficiency of photosystem II (Table 3). In control plants, the infection with *F. culmorum* resulted in the reduction of the photosynthesis rate (DR = 48.6%), the stomatal conductance (DR = 69.9%), the Ci/Ca intercellular to atmospheric ratio (DR = 8.45%), the quantum efficiency of Photosystem II (DR = 29.75%), the electron transport rate (DR = 25.15%), and the transpiration rate (DR = 53.03%). The impact of the infection in control plants was also characterized by an increase of the non-photochemical quenching (IR = 13.48%), and ABA content (IR = 25.11%). Under infected conditions, the seed coating with *M. guilliermondii* resulted in a lower reduction of the photosynthesis rate (DR = 30.99%), the stomatal conductance (DR = 42%), the quantum efficiency of Photosystem II (DR = 17.47%), the electron transport rate (DR = 5.49%), and a higher transpiration rate (3.41 mmol H_2_O m^−2^ s^−1^) (Table 3, Table 4). Jointly, it resulted in a lower increase of the non-photochemical quenching (IR = 9.20%), and a lower ABA content (0.181 ng/g FW). Unlike the plant control, the seed coating with *M. guilliermondii* increased the internal CO_2_ concentration (IR = 51.34%).

### 2.5. The Interrelationship among the Physiological Traits and the Disease Severity and Incidence

The correlation matrix (Figure 3) denotes that the traits that have significant positive correlations (*p* = 0.001) have been divided into two clusters based on the results of hierarchical clustering; cluster 1 include the disease severity, the anthocyanins and flavonoids contents and the non-photochemical quenching (r = 0.71–0.86), the second cluster includes the germination rate, the shoot length, the transpiration rate, chlorophyll content, the NBI index, the photosynthesis rate, the stomatal conductance, the plant biomass, phi2, ETR, the quantum efficiency of Photosystem II, and the photosynthetic electron transport (r = 0.19–0.97). In addition, all of the traits belonging to cluster 1 (disease severity, anthocyanins, flavonoids and the non-photochemical quenching) were negatively correlated to all of the traits belonging to cluster 2 (germination rate, shoot length, transpiration rate, chlorophyll, NBI, photosynthesis rate, stomatal conductance, plant biomass, phi2, ETR, quantum efficiency of Photosystem II, photosynthetic electron transport). The root length was positively correlated to ABA and negatively correlated to the intercellular to atmospheric CO_2_ (Ci/Ca). According to the correlation analyses, all the measured plant traits have been selected as potentially useful traits for investigating the implication of seed coating treatment in disease resistance of wheat plants.

## 3. Discussion

### 3.1. Effect of M. guilliermondii on Physiological Traits Under Non-Infected Conditions

Seed coating with beneficial microorganisms has been proposed as a very effective approach to promote durum wheat yield [8]. Previous studies demonstrated that treatment with *M. guilliermondii* improves maize (*Zea mays* L.) productivity and reduces the needs for chemical fertilization [20]. Besides, a combined treatment based on *M. guilliermondii* and a chemical fertilizer was found to promote tomato plant growth under control conditions [21]. Furthermore, recent studies showed that the strain *M. guilliermondii* Mg-11, isolated from Spanish vineyards, promoted Maize seedling development [22]. The observed positive effects of *M. guilliermondii* on seed germination and plant growth could be attributed to seed biopriming; the seed biopriming triggers the release and the production of phytohormones and enzymes which are involved in seed germination and plant growth ([23] Mahmood et al., 2016; [24] Bennett et al., 1998). Further, the growth-promoting potential of *Meyerozyma* spp. may also be accrued owing to its ability to induce IAA (indole-3-acetic acid) production [20]. The IAA hormone performs many regulatory functions, including stimulating plant cell enlargement, differentiation of xylem and phloem, cambium cell division, root initiation and lateral root formation [25].

Phytohormones play important roles in regulating development processes and signalling networks elaborated in plant responses to biotic and abiotic stresses [26]. Since ABA is frequently known as a stress hormone [27], the lower content of ABA in *M. guilliermondii* -treated plants, under both non-infected and infected conditions, denotes the healthy status of plants and the lower subjection to stressful conditions. This finding was in line with that of [15] who have observed a decrease in the ABA contents in plants of Maize inoculated with *M. caribbica*.

This study showed that in non-infected conditions, *M. guilliermondii* enhanced various photosynthetic attributes; namely chlorophyll content, NBI, the photosynthesis rate, the stomatal conductance, the Ci/Ca ratio, the electron transport rate, the transpiration rate, and resulted in a lower accumulation of flavonols and anthocyanins together with a lower non-photochemical quenching. Indeed, secondary metabolites including phenolic acids and flavonoids are necessary molecules for plant metabolism and growth, whereas the wide variety and high diversity of secondary products are key components for plants to interact with the environment in the adaptation to both biotic and abiotic stress condition [28]. The use of biostimulants has shown positive effects on plant growth [29], improving root hair and the N uptake, leading to increased net photosynthesis, transpiration rate, and the Ci/Ca ratio in maize [30].

### 3.2. Effect of the Infection with F. culmorum on the Physiological Traits in Control Plants

Despite the several existing studies on the disease development caused by *F. culmorum*, *F. graminearum* and *F. pseudograminearum* and the related host–pathogen interactions, very little is known about the impact of seed infection with *F. culmorum* on photosynthesis and the early host responses at the seedling stage [31]. Our study may contribute to improve the understanding of the impact of seed infection with *F. culmorum* on seedling stand, stomatal behaviour, and photosynthesis-related traits. As we observed in this study, the infection of seeds with *F. culmorum* induced a loss of seed germination resulting in a less dense plant stand. Moreover, the seedlings that managed to germinate from infected seeds had a reduced seedling growth. The infection effects are a consequence of pathogen dynamics; *F. culmorum* can effectively penetrate seedling roots, migrate from hypocotyl to the upper stem internodes and leaves, colonize the host’s tissue and cells, block the vascular bundles, and disturb nutritional supply, and metabolic processes [32,33], leading to a significant decrease in seedling growth. In our study, the infection effects were reflected in the observed continuous reduction of chlorophyll content which has been reported to result from either the destruction of chloroplasts and the gradual decomposition of chlorophyll in fungus affected cells [32], or from immobilization of nutrients required for chlorophyll formation and photosynthesis [34].

At the beginning of the disease process, plants induced flavonols as metabolites involved in the antioxidative reaction and defence response of plants [35]. However, along with the increasing severity and incidence of the disease, plants were no longer able to produce the antioxidant flavonols. As a result, the anthocyanins continued to rise which are involved in the reduction of photo-oxidative damage [36]. This resulted in the reduction of the NBI, which indicates that the plant directs its metabolism towards an increased production of flavonols (carbon-based secondary compounds), rather than towards its primary metabolism of protein synthesis (nitrogen-containing molecules) [37]. The latter state indicates the allocation of resources for the onset of defense reactions and biosynthesis of protective compounds rather than towards the process of photosynthesis, leading to a down-regulated photosynthesis [38].

The observed disturbed photosynthetic activity agrees with the findings of [39] who showed that the damage in the photosynthetic system occurred when the integrity of the host plant’s cellular structures was destroyed by fungal hyphae. [40] have mentioned that the mycotoxins produced by the fungi may also contribute to a strong reduction or even to the total loss of photosynthetic activity. The decrease in the photochemistry was associated with accumulation of ABA, stomatal closure and the related reduction of carbon content and transpiration. It can be suggested that the latter events could underline the disturbed water transport caused by the aptitude of the pathogen to colonize the root cortex and the vascular system [31]. The stomatal closure is an important factor contributing to low CO_2_ uptake and transportation of non-structural carbon which is an important component of photosynthesis leading to C starvation which further affects other processes [41].

### 3.3. Effect of M. guilliermondii on the Physiological Traits Following to Fusarium-Infection

Seed coating with *M. guilliermondii* lowered the damaging effect of *F. culmorum* on germination, and this is most likely owing to a repressed mycelium growth of *F. culmorum* on the surface of seeds. Following to the *Fusarium*-infection, the plants emerging from *M. guilliermondii*-coated seeds were able to reduce the development of wilt symptoms and to suppress the pathogen growth. These yeast species were characterized as active biological agents in several studies against fungal pathogens, including cabbage black leaf and tomato bacterial wilt disease [20], citrus blue mold [18], *Colletotrichum capsici* [42], and *Rhizopus stolonifer* in postharvest tomatoes [43]. Our research led the way to a possible antagonist potential of *M. guilliermondii* and competition for space and nutrients.

Moreover, our study clearly shows that applications of *M. guilliermondii* isolates could reduce pathogen damage and also promote durum wheat growth, leading to an increase in the shoots, roots length and the biomass compared to the control. This may be owing to the phytohormone produced by the yeast under stress conditions such as IAA and which involved in the plant growth. To face pathogens, many mechanisms activated by the yeast include the production of cell wall lytic enzymes and toxic volatile compounds [20], together with IAA production. In addition, some isolates are also known for their ability to induce systemic resistance against different pathogens. For example [18] reported that tomato fruits inoculated with *Pichia* (*Meyerozyma*, anamorph) *guilliermondii* activate defensive enzymes such as peroxidase polyphenoloxidase, superoxide dismutase, catalase, phenylalanine ammonia-lyase, chitinase and -1,3-glucanase activities [44].

Following the *Fusarium*-infection, the application of the bio-agent *M. guilliermondii* demonstrated an effective restoration of physiological traits and photosynthetic pigments compared to non-inoculated control of wheat plants, especially the increase of the photosynthesis rate. Moreover, the effect of *M. guilliermondii* on the leaf pigments of infected plants, causing a decrease in the contents of flavonoids and anthocyanins, together with an increase in the chlorophyll content, is only an illustration of the lower subjection of plants to stress and to the related oxidative damage. These results are in agreement with the finding of [15], where the inoculation of maize plants with *M. caribbica* promoted the chlorophyll contents under stress conditions.

In addition of the secondary metabolites, in plant–pathogen interactions, hormones modulate a series of defence responses, which limit pathogen spread [45]. In the present study, a decrease of synthesis of ABA was observed under *F. culmorum* infection in plants treated with *M. guilliermondii* compared the control. This finding was in agreement with the results of [15], who reported that Maize plant treated by *M. caribbica* showed lower endogenous ABA. Moreover, in Arabidopsis, ABA has been shown to antagonize the jasmonic acid signalling pathway, and this antagonism is considered responsible for the induction of disease susceptibility by ABA against the soilborne fungus *F. oxysporum* [46]. The observed potential of *M. guilliermondii* in growth promotion and disease control in seedlings might be achieved through the phenomenon so called bio-priming of seeds. In this context, [47] have reported that the colonization of seedling roots with biopriming microorganisms induces a broad-spectrum of resistance mechanisms in plants, reduces the root rot diseases caused by soil-borne plant pathogens.

## 4. Materials and Methods

### 4.1. Yeast and Pathogen Material

*M. guilliermondii,* a yeast strain INAT (MT731365) was collected from tomato stem isolated and characterized in the National Institute of Agronomy of Tunis, Tunisia. A pure culture was maintained at 4 °C on potato dextrose agar agar plates filled with 15 mL (PDA: extract of boiled potatoes, 200 mL dextrose, 20 g agar and distilled water, 800 mL). The active yeast cells were prepared in 250 mL flaks containing 50 mL of nutrient yeast dextrose (PDL, PDA without agar) with a loop yeast inoculum and incubated on rotary shaker at a speed of 180 rpm for 48h at 28 °C. After centrifugation (5000 rpm, 4 °C, 5min), the cells were collected and washed with distilled water. Then, the yeast cell concentration is adjusted to 108 cells m^−1^.

The fungal pathogen *F. culmorum* (FC) was isolated from brown lesions in wheat roots sampled from severely infected fields of north Tunisia, kindly provided by Dr. Samia Gargouri (National Institute of Agronomic Research of Tunisia). The pathogen inoculum used for plant inoculation were prepared as previously described by [11]. The fungal pathogen was maintained on PDA and was grown 7 days at 28 °C. Fungal macroconidia was produced in barley seeds. The seeds were inoculated by *Fusarium* strains and kept for 2 weeks at 25 °C in the dark. Conidial suspensions were diluted with autoclaved water to a final concentration of 1 × 106 conidia mL^−1^ containing 0.05% *v*/*v* Tween 20.

### 4.2. Seed Coating and Infection with F. culmorum

Seeds of durum wheat (*Triticum turgidum* subsp.durum.) cv. Karim, a sensitive soil-borne fungal pathogen variety, were used. Prior to use, seeds were surface sterilized; soaked for 2 min in an aqueous solution of 0.6% sodium hypochlorite (NaOCl), then for 2 min in 70% ethanol, and finally rinsed three times with sterile distilled water. The surface-sterilized seeds were subjected firstly to the infection with *F. culmorum*, and subsequently to the seed coating treatment with *M. guilliermondii*. In details, surface-sterilized seeds were soaked in the conidial suspension of *F. culmorum* or in sterile distilled water containing 0.05% Tween 20 as the non-infected control. The soaking seeds were kept for 16 h in the dark at 25 °C [8]. Afterwards, both infected and non-infected seeds were coated with *M. guilliermondii* filtrate as described by [8]. The coating product Agicote Rouge T17 (AEGILOPS Applications, France) was used, playing the role of the adhesion product, containing propane-1,2-diol (5–10%), polyethylene glycol mono (tristyrylphenyl) ether (5–10%), and 1,2-benzisothiasol 3(2H)-one (0.0357).

### 4.3. Wheat Growth and Experimental Conditions

All the experiments were conducted from October to mid December 2018 in the experimental facilities of the Faculty of Biology at the University of Barcelona (Spain). Wheat plants were grown in pots of 16 cm diameter containing a mixture of standard substrate: perlite (1:1; *v*/*v*) in an environmentally controlled growth chamber (Conviron E15; Controlled Environments, Winnipeg, MB, Canada). Pots were maintained under controlled conditions of temperature (22 ± 3 °C), relative humidity (40–50%), and a photoperiod of 16:8 h (light:darkness). Four seeds were planted per pot. The plants were uniformly irrigated every 2 days with 50% Hoagland‘s nutrient solution (Hoagland and Arnon, 1950). A Complete Randomized Design (CRD) with four combinations of seeds and three replicates was adopted (Figure 4). The four combinations of sown seeds consisted of: (i) non-infected and coated with *M. guilliermondii*, (ii) infected and coated with *M. guilliermondii* (iii) non-infected and non-coated control, and (iv) infected and non-coated control. Pots were rotated three times a week to ensure uniform growth conditions.

### 4.4. Measurements

The measurements were taken at different time points; at 7, 14, 21, 28, and 40 days after sowing (das) (Figure 4). They include measurements on the fully expanded leaf: Leaf gas exchange, leaf pigment content, seed germination, disease assessment, plant-growth traits, ABA analysis. Details are provided below (Figure 4).

### 4.5. Fusarium Crown Rot Disease Assessment

At 14, 21 and 28 das, three plants per treatment were sampled to measure the evolution of disease incidence and severity, according to the symptoms caused by the endophytic pathogen *F. culmorum* in wheat plants. The incidence index of the disease was recorded as the percentage of plants showing browning symptoms on stem base. The evaluation of the disease severity of FCR was scored in the seedling stage and rated on a 0–5 scale based on the typical symptoms of browning as previously described by [6].

### 4.6. Dualex Sensor Measurements and Pigments Analysis

At 14, 21, and 28 days, the contents in leaf photosynthetic pigments were measured in the last fully expanded leaves of 3 plants per treatment, using a portable leaf meter (Dualex Scientific+TM, FORCE-A, France) as described by Goulas et al. (2004). This optical sensor allows a non-destructive measurement of chlorophyll (Chl) content in leaves (given in µg cm^−2^), flavonols (Flav) and anthocyanins (Anth) contents in leaves (which are given in relative absorbance units), and the nitrogen balance index (NBI; ratio Chl/Flav).

### 4.7. Leaf Gas Exchange

Leaf gas exchange was measured in the fully expanded leaf at 28 das. Three plants per treatment were assessed using a LI-6400 infrared gas analyzer (LI-6400 Portable photosynthesis system, LI-COR, Lincoln, NE, USA), operating with a 6400-02 LED light source (LICOR) providing 400 μmol m^−2^ s^−1^ PPFD. The parameters measured were the rate of photosynthesis (Pr) (μmol CO_2_ m^−2^ s^−1^), the rate of transpiration (Tr) (mmol H_2_O m^−2^ s^−1^), the stomatal conductance (SC) (mol H_2_O m^−2^ s^−1^), the intercellular versus the atmospheric concentration of CO_2_ (Ci/Ca) and the photosynthetic electron transport rate (ETR) (µmol m^−2^ s^−1^).

### 4.8. ABA Extraction and Quantification

Abscisic acid (ABA) concentrations were measured at 28 das, by liquid chromatography coupled in tandem with modifications to mass spectrometry (HPLC-MS/MS). Thus, 50 mg samples of the last fully expended leaf were grounded by liquid Nitrogen. To each sample, 500 μL of isopropanol/methanol/acetic acid (90:9:1, *v*/*v*/*v*) extraction solvent was added, with 20 μL of deuterium-labelled abscisic acid (d6-ABA, 0.5 ppm, Saskatoon, Canada) as the internal standard. The obtained extracts were vortexed and centrifuged at 15,000 rpm at 4 °C for 4 min; the supernatants were collected, and the pellets were re-extracted with 200 μL of the extraction solvent and centrifuged again. Then, supernatants were pooled, dried with Rotavap for 120 min and reconstituted in 200 μL of methanol/water/acetic acid (90:20:0.01, *v*/*v*/*v*) vortexed, centrifuged (10,000 rpm, 10 min), and filtered through a 0.45mm PTFE filter (Waters, Milford, MA, USA). Finally, 5 mL of each sample was injected into the LC-MS/MS system. A calibration curve was created using serial dilutions of d6-ABA deuterium labelled internal standard (µg. g-1 FW).

### 4.9. Plant-Growth Traits

At 7 das, the percentage of the germinated seeds in pots was measured. At the end of the experiment (40 das), three plants per treatment were sampled for measuring the shoot and root length and plant biomass. For dry weight measurements, roots were washed in tap water until all substrate was removed, then root and shoot samples were dried at 70 °C for 48 h.

### 4.10. Statistical Analyses

The effects of the coating treatments and infection and their interaction on growth parameters, disease incidence and severity, and physiological parameters were determined through a two-factor (coating × infection) analysis of variance (ANOVA). The Least significant difference (LSD) test was used to assess differences between the treatments means. The Clustered Pearson correlation matrices were generated using the mean values of physiological and growth traits, which were jointed into most similar clusters according to the cluster function. All statistical analyses as well as figures for time course variation were built with RStudio 1.1.463.

## 5. Conclusions

It might be assumed that *M. guilliermondii* might have a pivotal function as antagonist and growth promoter. As an antagonist, it repressed the pathogen growth either through competing with it for space and nutrients, or through the release of antifungal compounds, these assumptions surely demand further investigations. In summary, our study provides now evidences, in this case for durum wheat a major crop in the Mediterranean basin, of the seed biopriming with microbial inoculants as a potential alternative to the use of chemical pesticides without any risk to humans, animals or the environment.

## Figures and Tables

**Figure 1 pathogens-10-00052-f001:**
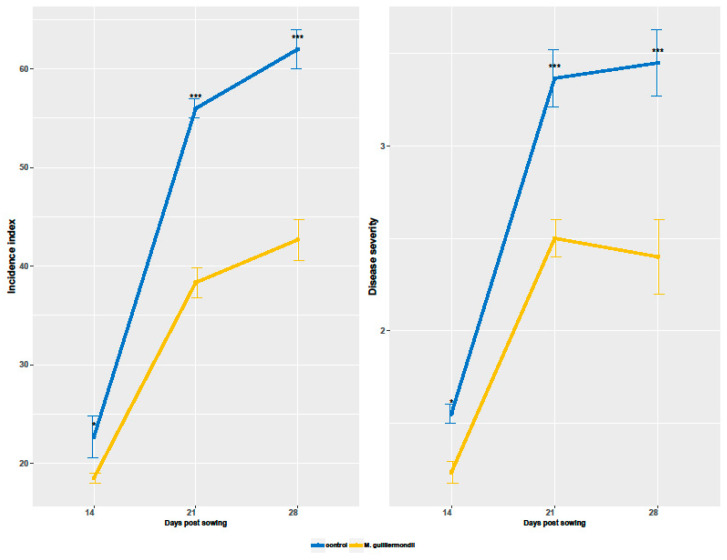
The effect of seed coating with *M. guilliermondii* on the evolution of *Fusarium* crown rot incidence and severity, at 14, 21 and 28 das. Data presented are the mean values ± standard deviation. The symbols of statistical significance are shown; ns: non-significant, *: *p* < 0.05; **: *p* < 0.01; ***: *p* < 0.001.

**Figure 2 pathogens-10-00052-f002:**
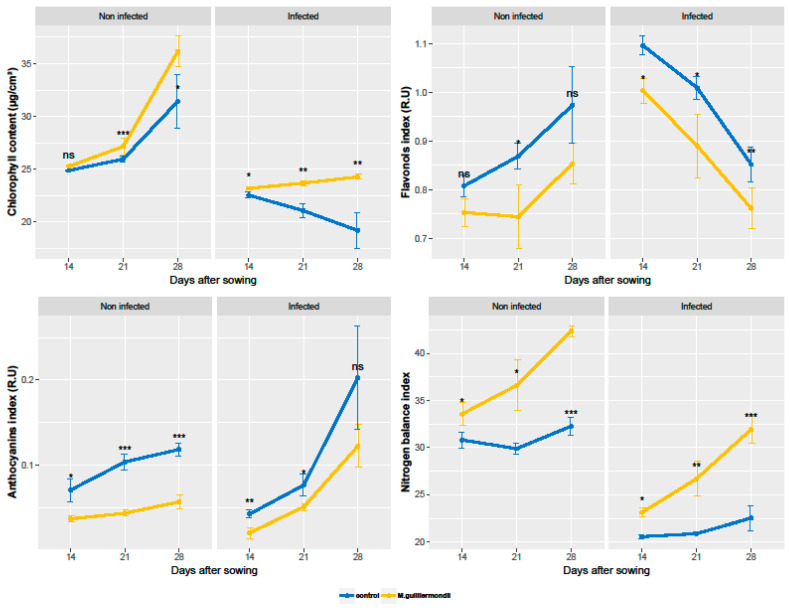
The effect of seed coating with *M. guilliermondii* on the kinetics of chlorophyll, flavonoids, anthocyanins, and nitrogen balance index in wheat leaves, under non-infected and infected conditions by *F. culmoum*. Data presented are the mean values with standard deviation. The symbols of statistical significance are shown; ns: non-significant, *: *p* < 0.05; **: *p* < 0.01; ***: *p* < 0.001.

**Figure 3 pathogens-10-00052-f003:**
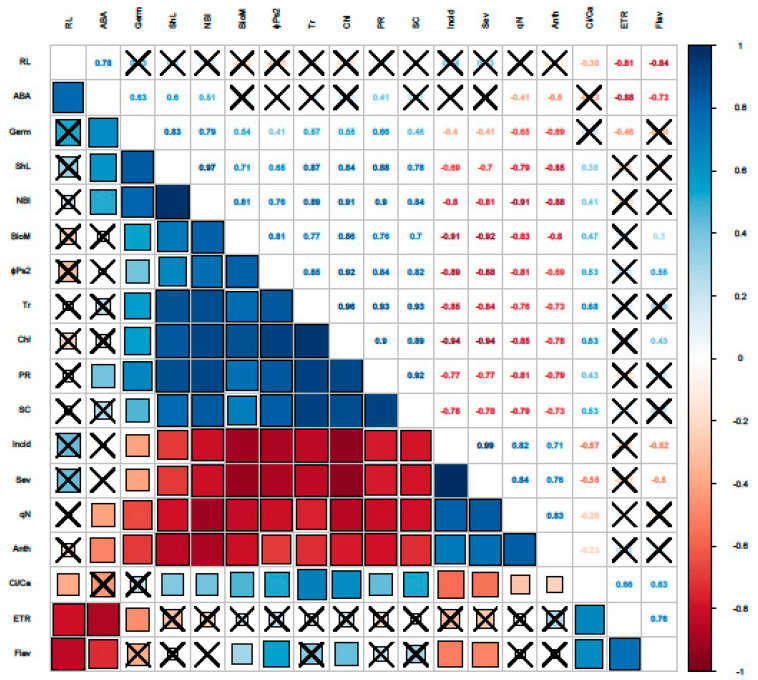
Pearson’s correlation matrix among physiological traits and disease incidence and severity. The darker, bigger blue and red squares (lower left diagonal) indicate stronger positive and negative correlations, respectively. Crossed cells indicate statistically insignificant correlations. Pearson’s coefficients (r) of correlation are shown (upper right diagonal). Abbreviations of metabolites are for Incid: incidence, Sev: severity, Anth: anthocyanins, qN: non photochemical quenching, Flav: flavonols, RL: roots length, ABA: ABA content, Ci/Ca: intercellular versus atmospheric CO_2_ concentration, Germ: germination, ShL: shoot lengths, Tr: transpiration rate, Chl: chlorophyll, NBI: nitrogen balance index, PR: photosynthesis rate, SC: stomatal conductance, BioM: Biomass, ɸPSII: quantum efficiency of Photosystem II, ETR: photosynthetic electron transport.

**Figure 4 pathogens-10-00052-f004:**
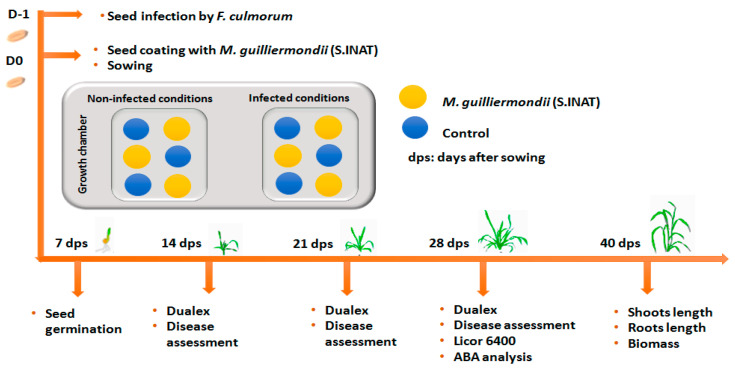
Schematic illustration of the experimental design.

**Table 1 pathogens-10-00052-t001:** Effect of seed coating with *M. guilliermondii* on wheat growth parameters, under non-infected and infected conditions by *F. culmoum*. Values are means ± SD of three replicates.

Traits		Germination %			Shoot Length (cm)			Root Length (cm)			Biomass (g)		
		Non-Infected	Infected	DR (%)	Non-Infected	Infected	DR (%)	Non-Infected	Infected	DR (%)	Non-Infected	Infected	DR (%)
Control		46.7 ± 9.5	26.7 ± 9.2	42.8	52.0 ± 1.6	35.5 ± 0.4	31.73	24.0 ± 0.2	15.9 ± 0.8	33.75	9.3 ± 05	4.6 ± 0.6	50.53
*M. guilliermondii*		93.3 ± 12.2	70.0 ± 9.4	24.97	73.6 ± 1.0	54.5 ± 0.4	25.95	40.6 ± 0.3	33.1 ± 0.5	18.47	10.8 ± 0.7	7.4 ± 0.7	31.48
ANOVA	Infection (I)	408 ns			181.7 ***			63.02 ***			27.90 ***		
	Coating (C)	3675 **			317.2 ***			130.02 ***			3.96 *		
	C × I	8 ns			4.9 ns			12.20 **			1.14 ns		

The sum square values with statistical significance are shown (ns: non-significant, *: *p* < 0.05; **: *p* < 0.01; ***: *p* < 0.001). DR: decrease from non-infected, SD: stander deviation.

**Table 2 pathogens-10-00052-t002:** Two-way ANOVA of chlorophyll, flavonols, anthocyanins, and the nitrogen balance index (NBI) contents.

	Chlorophyll(µg/cm²)	Flavonols (R.U)	Anthocyanins (R.U)	NBI (mg/g)
Coating(C)	53.5 ***	0.090 ***	0.0201 ***	348.6 ***
Infection (I)	339.6 ***	0.093 ***	0.0018 *	894.1 ***
C × I	102.0 ***	0.018 *	0.0427 ***	175.7 ***

The sum square values with statistical significance are shown (ns: non-significant, *: *p* < 0.05; **: *p* < 0.01; ***: *p* < 0.001).

**Table 3 pathogens-10-00052-t003:** Effect of the coating by *M. guilliermondii* and the infection by *F. culmorum* on the leaf gas exchange, photosynthesis, stomatal conductance (Sc), and ABA content, at 28 das. Values are means ± SD of three replicates.

Traits		Photosynthesis Rate (μmol CO_2_ m^−2^ s^−1^)		Stomatal Conductance (mol H_2_O m^−2^ s^−1^)		Intercellular to Atmospheric CO_2_ Concentration Ci/Ca		Quantum Efficiency of Photosystem II (ɸPSII)		Electron Transport Rate (ETR) (µmol m^−2^ s^−1^)		Transpiration Rate (mmol H_2_O m^−2^ s^−1^)		Non-Photochemical Quenching (qN)		ABA (ng/g FW)	
		Non-infected	Infected	Non-infected	Infected	Non- Infected	Infected	Non- Infected	Infected	Non-Infected	Infected	Non-infected	Infected	Non-infected	Infected	Non-infected	Infected
**Control**		18.30 ^b^ ± 3.20	9.40 ^c^ ± 0.25	0.399 ^b^ ± 0.01	0.12 ^c^ ± 0.01	287.66 ^ab^ ± 48.78	263.33 ^b^ ± 32.39	0.205 ^a^ ± 0.014	0.144 ^b^ ± 0.018	125722.5 ^b^ ± 733	94096.5 ^c^ ± 12567	4.94 ^b^ ± 0.99	2.32 ^c^ ± 0.53	1835.33 ^bc^ ± 78.38	2121.33 ^a^ ± 103.63	0.826 ^a^ ± 0.033	1.103 ^a^ ± 0.368
***M. guilliermondii***		25.26 ^a^ ± 5.10	17.43 ^b^ ± 0.73	0.66 ^a^ ± 0.24	0.24 ^bc^ ± 0.08	334.33 ^a^ ± 6.65	162.66 ^c^ ± 50.95	0.206 ^a^ ± 0.019	0.170 ^b^ ± 0.005	145665.0a ± 3059	137664.5^ab^ ± 5484	7.44 ^a^ ± 0.96	3.41 ^c^ ± 0.39	1729.33 ^c^ ± 37.58	1904.66 ^b^ ± 80.16	0.105 ^b^ ± 0.000	0.181 ^b^ ± 0.005
ANOVA	**Coating (C)**	168.75 **		0.1114 *		1519 ns		0.000520 ns		3.025e + 09 ***		9.67 **		78085 **		2.0240 ***	
	**Infection (I)**	210.00 **		0.3626 **		31519 **		0.006960 ***		1.178e + 09 **		33.17 ***		159621***		0.0938 ns	
	**C × I**	0.85 ns		0.0169 ns		18330 **		0.000469 ns		4.186e + 08 *		1.52 ns		9185 ns		0.0301 ns	

Values with different letter are significantly different at *p* = 0.05.

**Table 4 pathogens-10-00052-t004:** The decrease and the increase rate of the leaf gas exchange, photosynthesis, stomatal conductance (Sc), and ABA content at 28 das.

	Decrease Rate (%)						Increase Rate (%)	
Traits	Photosynthesis rate (μmol CO_2_ m^−2^ s^−1^)	Stomatal conductance (mol H_2_O m^−2^ s^−1^)	Intercellular to atmospheric CO_2_ concentration Ci/Ca	Quantum efficiency of Photosystem II (ɸPSII)	Electron transport rate (ETR) (µmol m^−2^ s^−1^)	Transpiration rate (mmol H_2_O m^−2^ s^−1^)	Non-photochemical quenching (qN)	ABA (ng/g FW)
Control	48.63	69.92	8.45	29.75	25.15	53.03	13.48	25.11
*M. guilliermondii*	30.99	42	51.34	17.47	5.49	54.16	9.20	41.98

## Data Availability

Data available in a publicly accessible repository.

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
