# Peer review of "Exploring the Potential of Meyerozyma guilliermondii on Physiological Performances and Defense Response against Fusarium Crown Rot on Durum Wheat"

_pathogens, 2021, doi:10.3390/pathogens10010052_

Round 1

Reviewer 1 Report

This article reports the biocontrol of Fusarium disease in durum wheat by using seeds coated with M. guilliermondii. In general, the article is relevant for the plant pathogen community and presents interesting results. However, the manuscript is a bit disorganized (mainly in table and figure numbering) and the English language has to be revised (the discussion section is clearly better than other sections of the manuscript).

Introduction

Line 24: please replace 30,4% by 30.4%

Line 27: please italicize F. culmorum

Line 39: please remove the word “through”

Line 42: please correct “seed coating agents”; “to improve emergence..”

Line 50: please define ABA and IAA, as well as other abbreviations, the first time you mention them in the manuscript

Line 54: correct “the key steps”

Line 58: italicize F. verticillioides

Line 59: This sentence needs a reference in the end

Line 62: italicize Rhizopus nigricans

Results

Line 70: affected positively or negatively?

Line 76: shouldn’t it be table 2?

Line 89: please replace , by .

Line 101: you should add the units of each measurement to the table

Table 3: This table is very difficult to read, you should divide this table into two, for example; define DR and IR in the table footer

Discussion

Line 182: promoted seed development of which plant?

Line 189: “cambium cell division” is repeated

Materials and Methods

Line 286: specify the quantity of PDA

Line 288: specify the rpm/g settings of the rotary shaker

Line 289: specify how long was the centrifugation step

Line 298: Why are the seeds infected first and only them coated with the yeast? Why not the other way around?

Line 299: subsp. Should not be italicized

Line 304: please italicize F. culmorum; there are several cases like this, please italicize all species names

Line 306: please correct the centigrade symbol and correct the reference style for Jaber, 2018

Line 307: The coating product was used as a medium for the yeast? This is not clear

Line 315: please correct the centigrade symbol

Line 319 and 327: Shouldn’t it be figure 4 instead of 1?

Reviewer 2 Report

The manuscript discusses the potential of a new Meyerozyma guillermondii strain for the control of Fusarium Crown rot in durum wheat. In addition to disease control, the seed treatment has to improvement growth and some parameters linked to the well-being of crop. The experimentation is interesting, clear, well done and well presented. I suggest, in a future paper, to evaluate the strain also with regard to a possible reduction of incidence and severity of Fusarium head blight and mycotoxin content (the main problem in Italian soft and durum wheat). Since I have not identified any particular concern to be solved, I consider the manuscript worthy of publication.
